# Assessment of the Efficiency of Measuring Foot and Ankle Edema with a 3D Portable Scanner

**DOI:** 10.3390/bioengineering10050549

**Published:** 2023-05-03

**Authors:** Julien Beldame, Riccardo Sacco, Marie-Aude Munoz, Marion Masse, Matthieu Lalevée

**Affiliations:** 1ICP-Clinique Blomet, 136 bis Rue Blomet, 75015 Paris, France; 2Clinique Megival, 1328 Avenue de la Maison Blanche, 76550 Saint Aubin sur Scie, France; 3Department of Orthopedic Surgery, Rouen University Hospital, 37 Bd Gambetta, 76000 Rouen, France; 4Centre Médical Achille, 200 Avenue des Prés d’Arènes, 34070 Montpellier, France; 5CKS, Centre kiné sport Dieppe, 32 Rue Louis Blériot, Neuville les Dieppe, 76370 Dieppe, France; 6CETAPS EA3832, Research Center for Sports and Athletic Activities Transformations, University of Rouen Normandy, 76821 Mont-Saint-Aignan, France

**Keywords:** optical scanner, 3D imaging, foot volume, volumetric measurements, water volumeter

## Abstract

**Background:** To prospectively evaluate the reliability of a portable optical scanner compared to the water displacement technique for volumetric measurements of the foot and ankle and to compare the acquisition time associated with these two methods. **Methods:** Foot volume was measured in 29 healthy volunteers (58 feet, 24 females and 5 males) by a 3D scanner (UPOD-S 3D Laser Full-Foot Scanner®) and by water displacement volumetry. Measurements were performed on both feet, up to a height of 10 cm above the ground. The acquisition time for each method was evaluated. The Kolmogorov-Smirnov test, Lin’s Concordance Correlation Coefficient, and a Student’s *t*-test were performed. **Results:** Mean foot volume was 869.7 +/− 165.1 cm^3^ (3D scanner) versus 867.9 +/− 155.4 cm^3^ (water-displacement volumetry) (*p* < 10^−5^). The concordance of measurements was 0.93, indicative of a high correlation between the two techniques. Volumes were 47.8 cm^3^ lower when using the 3D scanner versus water volumetry. After statistically correcting this underestimation, the concordance was improved (0.98, residual bias = −0.03 +/− 35.1 cm^3^). The mean examination time was 4.2 +/− 1.7 min (3D optical scanner) versus 11.1 +/− 2.9 min (water volumeter) (*p* < 10^−4^). **Conclusions:** Ankle/foot volumetric measurements performed using this portable 3D scanner are reliable and fast and can be used in clinical practice and research.

## 1. Introduction

Edema is the swelling of soft tissues due to an abnormal buildup of interstitial fluid. It is caused by the increased movement of fluids from the intravascular space towards the interstitial space or decreased movement of water from the interstitial tissue to the capillaries or lymphatic vessels. At the level of the ankle and the foot, the edema severity is compounded due to orthostatic pressure, making this joint segment a predominant location, which generates significant symptomatology (pain, tension, and joint stiffness), difficulty in putting on one’s shoes (a source of loss of locomotor function), and significant psychosocial impact [1,2].

The circulatory system, lymphatic system, and kidneys are the main body systems that assist in preserving the proper fluid balance in the body. Any disruption of the homeostasis of these systems can lead to fluid retention. Peripheral edema, which results from the buildup of fluid in the tissue, affects the legs and feet. In the case of heart failure decompensation, for instance, the clinically significant size of volume change for edema detection is 13.1% [3]. Therefore, the recognition of more subtle volume variations is useful in clinical practice, such as in the clinical setting of deep vein thrombosis, compartment syndrome, chronic venous insufficiency, and medications associated with edema (calcium channel blockers and other vasodilators). Moreover, previous evidence demonstrated that a normal patient’s uninjured foot and ankle can reliably be used as a control limb at any time of day, whether the subject is ambulating or not [4]. They came to the conclusion that a normal patient’s uninjured foot and ankle could be used as a control limb at any time of day, whether the subject was ambulating or not. It is possible to accurately compare the volume of the affected side to the volume of the injured foot. Volumetric measurements are recommended for both bilateral and one-sided traumatic foot and ankle swellings.

Foot and ankle volume measurements have many applications. The most obvious are in the medical field. The use of a device to assess the volume of the foot and ankle would be useful in assessing and monitoring several pathologies. For instance, in the case of heart failure, the assessment of foot and ankle edema variation could represent a complementary and indirect aid for the physician in evaluating the effect of medical treatment.

Lymphatic issues can also be monitored using this same method [5,6,7]. It would also be useful to monitor edema after surgery on the lower limbs. Most of the time, after surgical treatment, the edema gradually regresses until it disappears completely. Monitoring the edema would allow the physician to know the patient’s stage of improvement by following them regularly in the clinic. Similarly, volume monitoring would allow comparison of different rehabilitation methods and techniques, such as drainage massage, compression stocks, or bandages, after surgery [8]. On the other hand, if there is a progressive worsening of the volume after surgery associated with pain or other alarming signs and symptoms, the physician should promptly consider and rule out potential complications such as deep vein thrombosis or infection [9].

The manufacturing of shoes or insoles could also benefit from this type of technology. Knowing the exact volume, the difference in volume between the genders, the variation in volume over the course of a day, or sports activities would be beneficial for the manufacture of shoes [10,11,12,13]. This technology could be applied in the future to assess whether footwear for cavovarus foot, progressive collapsing foot deformity, or hallux valgus is associated with subtle improvements in foot and ankle edema [14,15,16,17]. Sports footwear enhancement can also benefit from foot and ankle volume measurements with two goals: to decrease the incidence of injury and to improve sports performance [18,19].

Having reliable methods to assess foot and ankle edema seems essential. In the literature, several methods for measuring lower limb edema have been described:Measuring foot and ankle volume by water displacement, or water volumetry, remains the reference method [20,21,22]. Some authors use inverse water volumetry [23], which consists of placing a dry foot in a volumeter that has been filled up to a predetermined level. Foot volume is determined either by assessing the volume of water that overflows from the foot volumeter or by the volume of water that needs to be added after retracting the foot from the volumeter to return to initial water levels (“inverse” method). The advantages of water volumetry are practicability and reproducibility [22]. However, in daily clinical practice, managing water volumes, maintaining water hygiene, and taking time may represent significant issues. Moreover, immersing a limb with any kind of skin lesion in water is not advised.In daily clinical practice, perimetric (non-volumetric) measurements constitute the most frequently adopted method, although reproducibility and inter-rater reliability are low [24]. To improve reproducibility, several studies have shown an interest in figure-of-eight methods [25] or the added value of professional experience in raters [26]. Using tape measure methods, some studies have attempted to calculate the volume of a limb based on mathematical methods, without, however, paying attention to distal volumes (fingers or toes) [26], and with 8 to 12% error margins when compared to the reference method [27]. For foot/ankle measurements, some surveys have proposed mathematical formulas to determine volume from perimetric measurements performed on particular cutaneous points of reference [28].Several 3D scanning measurement methods have been described in previous surveys. Most of them were used for knee joint measurements. Of note, knee joint 3D morphology is less problematic to assess than that of the foot and toes [29,30]. Indeed, the foot as well as the hand, because of the difficulties in defining precisely the volumes of the fingers and toes, are more difficult to assess [31]. These techniques are associated with high reproducibility. However, they have often been assessed with reference to tape measure methods [32], much less the reference method (water volumetry) [27]. Tape measures are non-weight-bearing measurements. Measuring foot and ankle volume without the application of weight and only on limb segments, excluding the foot and toes, is potentially biased [33].

In both clinical and research contexts, it is critical to have a reliable and simple method for determining foot volume when it comes to evaluating peripheral edema or certain types of foot and ankle injuries [34]. While “pitting” in clinical settings and the use of water volumetry or tape measurements (either ankle circumference or figure of eight) in research settings are considered the gold standard procedures, these techniques are prone to human error in measurements.

Thus, our primary objective was to assess the reliability of a 3D portable scanner for obtaining volumetric measurements of the foot and ankle in comparison with water volumetry, considered the reference method. A secondary objective was to compare the time taken by both methods.

We hypothesized that this portable 3D scanner was reliable and allowed quick volumetric measurements of the foot and ankle.

## 2. Materials and Methods

This prospective, non-interventional study was conducted on 29 healthy volunteer subjects (58 feet assessed overall). Oral consent was obtained before participation, and approval for this study has been obtained beforehand from the institutional review board of CPP GHT Grand Paris Nord Est on 29 September 2021 (approvals # Si-RIPH2G: 21.01741.000023 and NRCB 2021-A01802.39). Subjects who declined to participate in the study, presented with dermatological disease, or were unable to stand on both feet were not included. Overall, 24 female and 5 male subjects were included (mean age 35.6 +/− 9.5 years, range 9–55). Self-declared shoe size was 38.17 +/− 3.23 (range 30–45). Demographic characteristics are shown in Table 1. Each foot was measured both by a portable optical scanner and by water volumetrics. 

A non-irradiating, portable (13 Kg, 27 × 52 × 22 cm) 3D scanner was used (UPOD-S 3D Laser Full-Foot Scanner^®^, East Lake, Wuhan City, Hubei Province, China 430075). It allowed for the acquisition of 3D models of foot/ankle up to a maximum height of 11.5 cm from plantar support. The scanner was used with its dedicated software (UPOD-3D Foot full scan, East Lake, Wuhan City, Hubei Province, China 430075), allowing to obtain the 3D models of the foot (linear precision 1 mm.) and to automatically export a series of specific foot/ankle measurements (perimeters, distances, angles). Subjects were positioned standing on both feet (one foot in the scanner, the other on a footrest at the same height). Each foot was scanned in less than 4 seconds. (Figure 1). The measurement method used as a reference was water displacement with a volumeter (overflow technique) [35]. (Figure 2).

Each foot of each subject was first immersed in the volumeter to determine its volume. After the volume of each foot had been determined by the overflow method, the volume of each foot was then determined by a 3D optical scan, followed by computerized processing of the measurements. We elected to limit measurement height to 10 cm above plantar support and to focus on the foot and ankle volume only, excluding the distal leg, to minimize any possible bias associated with the angular positioning of the ankle in the scanner. All measurements were carried out by the same rater. The mean time of acquisition for each foot was obtained by an independent rater, including removal of shoes, drying of the feet, and return to the initial “shoes on” status.

Statistical analyses were performed by an independent statistician. Concordance between the two methods was validated using the Bland-Altman method and by Lin’s (CCC), after conducting a Kolmogorov-Smirnov test to ensure that differences between the two methods were normal. As for acquisition times for each foot, durations of clinical tests were studied by assessing the differences between mean spent times (paired quantitative data measured in the same individuals), using the Student’s *t*-test (threshold 5%), after verifying our hypothesis of normalcy using the Kolmogorov-Smirnov test.

## 3. Results

The mean foot volume when measured by a 3D scanner was 821.9 +/− 162.6 cm^3^, versus 869.5 +/− 160.0 cm^3^ when using water-displacement volumetry (*p* < 10^−5^). The concordance of gross measurements, measured by Lin’s CCC, was 0.93, indicative of an excellent correlation between the two techniques. No deviation from normalcy was shown for the difference in measurement between water volume and scanner volume (*p* = 0.2), which allowed the application of the Bland and Altman method (Figure 3);The measurement discrepancy was 47.8 cm^3^, showing underestimation when using a 3D scanner versus water volumetry. After correcting the results yielded by the 3D scanner method for this value (“corrected 3D scanner measurement, in which the measurement discrepancy of 47.8 cm3 was considered and normalized in the comparison of volume measurement between the two methods);An excellent concordance was demonstrated between the two techniques. After statistically correcting the 3D scanner volume underestimation, compared with the water volumetry, the concordance was improved from 0.93 to 0.98 (LIN’s CCC = 0.98, residual bias = −0.03 +/− 35.1 cm^3^), as shown in Figure 4. The mean examination time was 4.2 +/− 1.7 min when using the 3D optical scanner versus 11.1 +/− 2.9 min when using the water volumeter. This was a statistically significant difference (*p* < 10^−4^), as shown in Table 2.

## 4. Discussion

Our study confirmed that our 3D optical scanner achieved excellent correlation with the reference method (0.92 in gross measurements, 0.98 after correcting for an underestimation bias when using the scanner method). Further, this technique allows for a significantly reduced time spent on examination when using water volumetry. Therefore, our hypothesis was confirmed. To the best of our knowledge, this is the first study that reports high correlation rates between a laser-based measuring technique and the reference method.

Several multidimensional morphological parameters, including foot length, width, circumference, and navicular height, characterize the volume and structure of the human foot. Foot and ankle volume are subjected to changes secondary to aerobic activities, lower-extremity trauma, or pathologic conditions [2,4]. The change following surgical treatment is also important to consider [34], especially since the technique of the water basin, because of the wounds, is not suitable. The use of 3D optical scanners to track foot and ankle volume following surgery would also be an improvement. For example, it would be possible to detect septic complications following surgeries through such monitoring, although many advances in this area have reduced the incidence of postoperative infections [9].

In the clinical setting, volume measurements of the foot and ankle are performed to assess the severity of peripheral edema and the outcomes of medical therapies. In order to measure foot and ankle volumes, both 3D optical scanners and water displacement procedures are highly reliable. Water displacement or water volumetrics are considered the gold standard. To date, several measurement methods and geometric algorithms have been investigated, including the prism approximation, the figure-of-eight tape measurement, and size measurement with the Brannock device [3,6,25,26]. These methods have been implemented as a useful alternative to water displacement measurements, but they are subject to the impact of human factors on measurement errors. According to our results, 3D optical scanners could be considered the new gold standard.

Given that several diseases related to changes in foot volume are chronic disorders that may benefit from long-term surveillance, such as chronic venous leg ulcers [7], the prospects of applying this technology in clinical practice are attractive. This technology should also be applied in clinical trials to obtain reliable, non-invasive, and objective measurements of foot and ankle volume.

In the literature, surveys assessing limb measurement with the use of a 3D technique are relatively recent and have often been spurred by commercial entities interested in updating anthropometric data on the general population, notably in relation to clothing or footwear [36,37,38]. In the field of scanners, laser technologies remain the gold standard (even though more affordable optical and infrared scanners have been used at times) [39]. These innovative methods allow for any contact with the limb under study, thereby avoiding skin-related contraindications and the handling of water.

Several full-limb scanning series have reported positive results (most of them have been concerned with the knee) [29,30]. Series that specifically address distal features in lower limbs have been scarce because such studies need to cover multiple, contiguous segments and low volumes (toes):-The Volumeter^®^ (Bosl Medizintechnik, Aachen, Germany) only allows infra-red, off-load optical measurement from the malleoli and on the most proximal 36 cm [40]. The correlation with the reference method is excellent, but perimetric measurements need to be performed.-The Artec Eva^®^ optical scanner described in Hofmann et al. [41] can only be used in off-load situations. So far, its measurements have only been validated in comparison with figure-of-eight methods.-The Perometer^®^ has been used in several studies with excellent reproducibility. However, it is an infrared device (not a laser one) and has often been compared with tape measure methods [32], much more rarely with the reference method. Only Tierney et al. compared it with the reference method, obtaining a correlation of 97% [27]. However, this study focused on the leg segment, excluding the ankle and foot, in which, due to the presence of the toes and a very variable topographic anatomy, volumetric measurements are known to be difficult.-Our 3D laser scanner and its associated technology are not novel. However, when cited in previous surveys, it was used to measure the length, width, and circumference of the foot but never for volumetric measurements [42,43]. It has been used in large-scale measurement series, notably for anthropometric measurements of the foot, or as an alternate method to molding or ink/manual impression taking for manufacturing orthopedic footwear or plantar orthotics [44,45].

Few studies have investigated the time needed for the acquisition of the different techniques available for measuring foot and ankle volume. De Vrieze et al. [46] reported that 30 min per patient were needed in five upper-limb edema measurement techniques (notably a mean duration of 4 minutes when using water measurements, significantly higher than the time required in our scanning technique). Devoogdt et al. [35] found a mean time of more than three minutes per foot when using the water volumeter (a figure significantly higher than in perimetric measurements). In our study, we found a relatively similar overall duration of more than 10 min (for both feet) when using water volumetry, which reflects the low practicability of this technique. Conversely, the time needed for conducting a scanner-based examination (including removal of shoes, subsequent return to a “shoes-on” situation, and computer processing of data) was around two minutes [35,46], making it an ideal technique in clinical practice.

Our sample of 29 subjects (58 feet/ankles) is broader than most samples studied in previous surveys (30 subjects in Labs et al. [40], 20 limbs for the validation of the Perometer^®^ [10]; Tierney et al. [27] reported on 20 leg segments). Furthermore, the validation of our scanner was carried out within a population in which very varied volumes were observed, ranging from 537.3 cm^3^ to 1217.3 cm^3^, thus encompassing feet ranging from sizes 30 to 45.

This 3D optical scanner achieved excellent correlation with the reference method: 0.92 in gross measurements and 0.98 after correcting for an underestimation bias when using the scanner method. A measurement bias was observed in our study. However, it was lower than that observed in Tierney et al. [27] (7% for Tierney et al. vs. around 5% according to our method). The measurement discrepancy was 47.8 cm^3^, showing underestimation when using a 3D scanner versus water volumetrics. The discrepancy is thought to be caused by the phenomenon called siphonage, which resulted in an overestimation of the foot and ankle volume measured by the water volumetry compared to the 3D scanner. After correcting the results yielded by the 3D scanner method for this value (“corrected 3D scanner measurement”, in which the measurement discrepancy of 47.8 cm^3^ was considered in the comparison of volume measurement between the two methods), an excellent concordance was demonstrated between the two techniques (LIN’s CCC = 0.98, residual bias = −0.027 cm^3^ +/− 35.10 cm^3^), as shown in Figure 4. Concordance between the two methods was validated using the Bland-Altman plot to demonstrate trends and systematic errors and the intra-class correlation coefficient to establish the precision of the measurements. Lin’s method (CCC), after conducting a Kolmogorov-Smirnov test, ensured that the differences between the two methods were normal. This discrepancy between the measurements of the two methods is thought to be secondary to the phenomenon called siphonage. The unbalanced forces caused by the difference in height between the inlet and outlet under atmospheric pressure, create “siphoning”: the physical phenomenon of the fluid moving from a lower to a higher and then to a lower level [47,48]. This issue is of particular relevance in the context of an increasing shortage of water resources and the metering and measurement of water in engineering channel communications, but it has not been studied in the water volumetry for the evaluation of foot and ankle volumes. According to the laws of hydraulics, the movement of water to determine the flow rate in pressure flows is sufficient to measure the speed of the water. The cross-sectional area is usually known and limited by the walls of the conduit. The flow rate is determined by multiplying the fluid flow rate by the living cross-sectional area of the flow. Thus, the optimal siphon pipe should have a minimum value of hydraulic resistance and a maximum water-giving capacity [48]. The 3D optical scanner obviates the issue of determining the actual flow rate and water distribution by the siphon pipe. To date, no previous study has highlighted this particular measurement problem of the water volume for the evaluation of foot and ankle volumes.

In the medical field, the use of a 3D optical scanner would allow the monitoring of many pathologies quickly and without the drawbacks of the water displacement volumetry method. For example, monitoring a lymphatic drainage problem or a cardiac decompensation by measuring edema with the water displacement volumetry technique can be dangerous for patients since most of them are exposed to skin complications such as arterial or venous ulcers [34]. Postoperative edema is also one of the most difficult to assess. Immersing the foot in a volume of water is absolutely not appropriate in this condition, which has a high risk of infectious complications [9]. In all these medical applications, the assessment of the foot and ankle volume with a 3D optical scanner would be perfectly appropriate.

The high speed of the 3D optical scanner and its respect for hygiene would also allow a large number of subjects to be scanned in order to create corridors of normality to improve the manufacture of shoes. Knowing these normality corridors would help build a standard shoe model that would suit most of the population. On the other hand, if a specific construct is needed for a foot and ankle deformity such as flatfoot, recently renamed progressive collapsed foot deformity, this would also be possible [14].

Also, in the field of sports, the analysis of the foot morphotypes of the athletes using this device could make it possible to improve the performance of the shoes while avoiding the most common injuries. Some of them are due to an inappropriate distribution of the volume of the shoes or a bad adaptation to the foot variations during exercise [49,50,51].

Our study was not without limitations. First, volumetric measurements were performed on a healthy population. It is thus likely that the reproducibility of optical measurements may be affected in patients unable to stand on both feet (causing asymmetric plantar compression) or presenting overlapping toes in the context of forefoot deformity. Further study is needed on these aspects. Moreover, we did not perform a priori power calculations, which could limit the interpretability of our results. However, as mentioned above, our study included a significant number of feet compared to previously published studies on the same topic. It is important to note that our device measures changes in volume in all soft tissues (muscle and fat) and not just the variations of peripheral edema in the foot and ankle. Third, although any novelty in the methodology has been described compared to previous studies on this subject, we presented interesting results such as the siphonage phenomenon as a potential cause for the measurement discrepancy between water volumetric and 3D scanner measurements, with possible applications in future studies. Finally, this type of portable scanner remains expensive (7000 to 8000 euros) and is currently not widespread in clinical practice, thereby limiting the reproducibility of this study. The investment and operating costs of any new technology should be considered in parallel with improved usability and measurement accuracy. This current technology appears to present a path towards a substantial improvement compared to other instrumental methods of measuring foot and ankle volume. In the future, portable 3D optical scanners could be used both in clinical settings and in patients’ homes. 

## 5. Conclusions

Our portable 3D laser scanner showed an excellent correlation with the reference method and significantly reduced examination time. Its portable use, its speed, and the absence of contraindications linked to immersion in water make it an ideal clinical or research tool for measuring and monitoring foot and ankle edema.

## Figures and Tables

**Figure 1 bioengineering-10-00549-f001:**
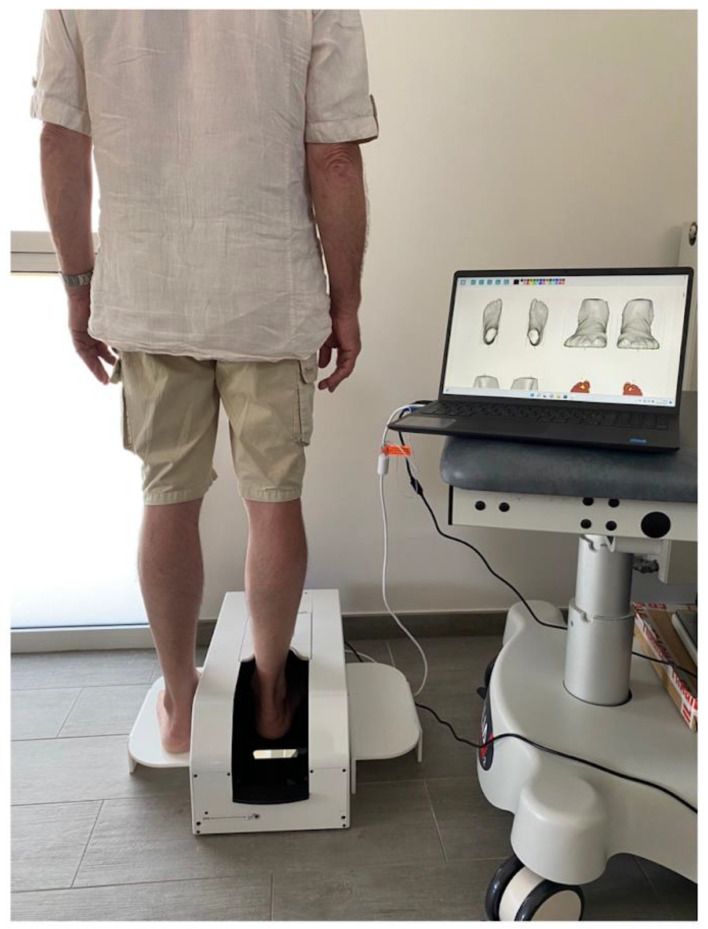
Subjects were scanned in a standing position (one foot in the scanner, the other one on a footrest at the same height).

**Figure 2 bioengineering-10-00549-f002:**
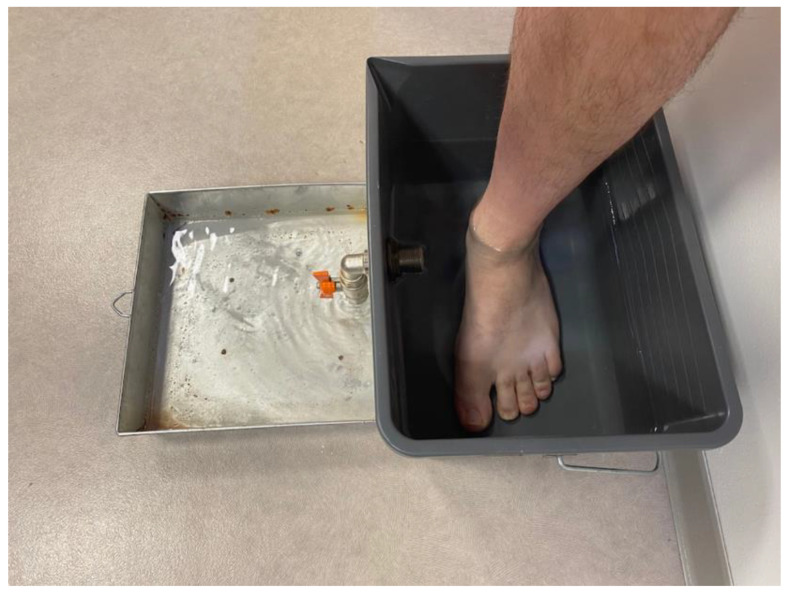
A volumeter was set up with a tap allowing overflow. After filling above the level of the tap, the latter was opened to obtain the reference level (after a few minutes in order to obtain flat, stable water). The feet of the subjects were slowly immersed until both feet stood firmly on the ground, causing the displacement of a volume of liquid outside the volumeter through the tap. The liquid corresponding to the volume of the submerged foot was then collected and weighed.

**Figure 3 bioengineering-10-00549-f003:**
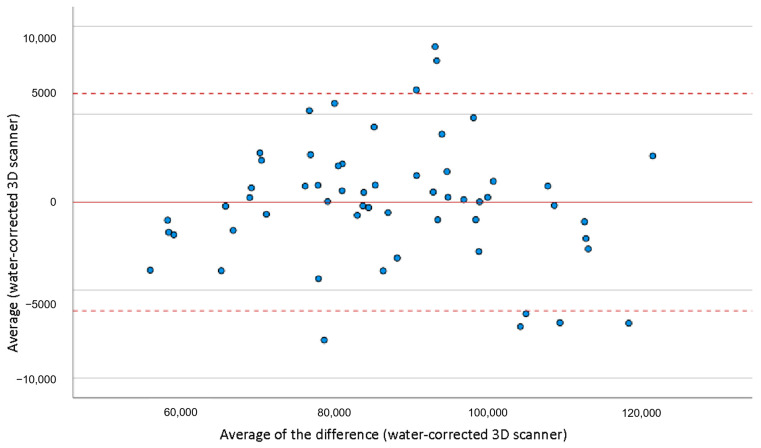
Bland-Altman representation.

**Figure 4 bioengineering-10-00549-f004:**
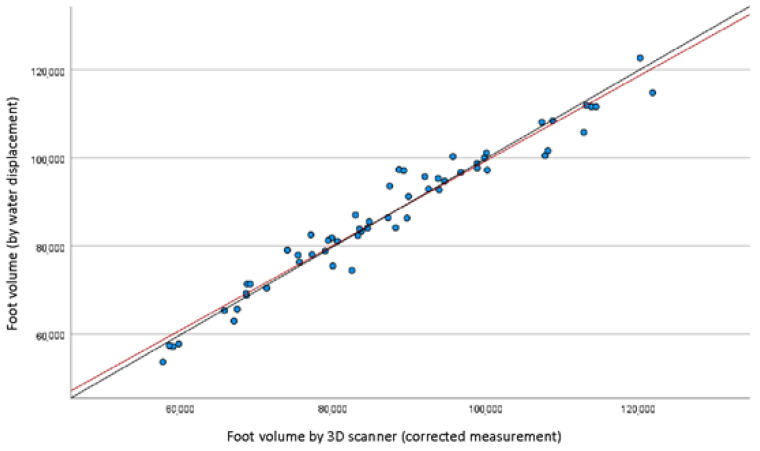
Volume correlation diagram between the water displacement and the 3D corrected scanner measurements. The red line represents the linear regression curve and the black line represents the egality line of the two techniques.

**Table 1 bioengineering-10-00549-t001:** Demographic characteristics of the population.

Population	n = 29
Gender (Male/female)	5/24
Age (years) average +/− standard deviation Minimum age (years) Maximum age (years)	35.6 +/− 9.5 9 55
Shoe size (European) Average +/− standard deviation Minimum Maximum	38.2 +/− 3.2 30 45

**Table 2 bioengineering-10-00549-t002:** Volume measurement according to methods.

Volume (cm^3^)	3D Scanner	Water Displacement	*p*
Average +/− Standard Deviation	821.9 +/− 162.6	869.5 +/− 160.0	<10^−3^
Minimum	528.2	537.3	
Maximum	1169.5	1227.4	
**Measurement time (min)**	4.2 +/− 1.7	11.1 +/− 2.9	<10^−3^

## Data Availability

Research data is not publicly archived and is available on request from the corresponding author.

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
