# Peer review of "Assessment of the Efficiency of Measuring Foot and Ankle Edema with a 3D Portable Scanner"

_bioengineering, 2023, doi:10.3390/bioengineering10050549_

Round 1

Reviewer 1 Report

How about adding the word "efficiency" to the title?

Can you give a few more examples like the ones mentioned in lines 41-43?

The contents of lines 44-46 and 48-49 are the same, so omit them or express them differently.

In lines 55-57, it is unreasonable to evaluate the degree of cardiac decompensation only by foot & ankle edema in the medical field. Edema does not directly arise for that reason alone. Relax the expression, such as "It may help indirectly."

As in line 66, "foot and ankle volume" will be helpful in identifying changes in the patient's condition after surgery, but it is unreasonable to know 'post op infection'.

I do not agree that 'lines 72 - 74' evaluate bone deformity by edema measurement.

Reviewer 2 Report

Although this manuscript shown plenty of interesting results for volumetric measurements of the foot and ankle using a commercial available 3D scanner, the novelty and general significance of this work was weak from the respective of the readers because research papers should at least report novel methods or materials for the interest of the readers.

In this manuscript, the authors reported plenty of interesting results for volumetric measurements of the foot and ankle using a commercial available 3D scanner. The topic is relevant in the bioengineering field. The author given the conclusion that the data obtained from a 3D laser scanner was consistent with the reference method, and examination time was reduced. The conclusions are consistent with the data presented in the manuscript. The references are appropriate.

1.Regarding the methodology, novel methods or materials should be included in a research paper to contribute to the related subject area.

2. Figures and tables can be improved from the perspectives of format and quality.

The quality of English language is fine.

Reviewer 3 Report

Authors present comparison of two methods for measuirng the ankle volume, one in the standarised form and and with tthe 3d scanner. 

I have few concers regarding the statistical analysis of the results.

Why authors used Student t-test for paired comaparions?

Student t-test is used for two groups, but for the same group measued in two different time point you shouls use paired t-test. 

What is “corrected 3D scanner measurement”?

This has to be better explained in the manuscript. 

In the table 2, the volume measures look quite similar, but how is possible that the p-value is <0.00001? 

That means that there is a statistically significant difference between the results of the two methods? This has to be clarified.

Results shoulde be presented more infomrative to the reader, it is hard to see thier importance for the method, escpecially in the abstract part. 

Round 2

Reviewer 2 Report

I have no further comments.

Minor editing of English language required.

Reviewer 3 Report

Dear authors, thank you fro your corrections and modifications.